# Image Compression Is an Effective Objective for Visual Representation Learning

## Abstract

Self-supervised pre-training is an effective method for initializing the weights of vision transformers. In this paper, we advocate for a novel learning objective that trains the target model to use a minimal number of tokens to reconstruct images. Compared to the existing approaches including contrastive learning (CL) and masked image modeling (MIM), our formulation not only offers a new perspective of visual pre-training from the information theory, but also alleviates the degradation dilemma which may lead to instability. The idea is implemented using Semantic Merging and Reconstruction (**SMR**). SMR feeds the entire image (without any degradation) into the target model, gradually reduces the number of tokens throughout the encoder, and requires the decoder to maximally recover the original image in the semantic space using the remaining tokens. We establish SMR upon the vanilla ViT and two of its variants. Under the standard evaluation protocol, SMR shows favorable performance in visual pre-training and various downstream tasks. Additionally, SMR enjoys reduced pre-training time and memory consumption and thus is scalable to pre-train very large vision models. Code is submitted as supplementary material and will be open-sourced.

## 1 Introduction

Self-supervised visual pre-training plays a fundamental role in computer vision and it is particularly important to initialize the state-of-the-art vision transformers (Dosovitskiy et al., 2021; Liu et al., 2021). Recent years have witnessed a rapid development of visual pre-training. The popular algorithms include contrastive learning (CL) (Chen et al., 2020; He et al., 2020) and masked image modeling (MIM) (Bao et al., 2022; He et al., 2022; Xie et al., 2022). These approaches have shown the ability to learn representations from an image dataset without semantic labels and transfer the knowledge to various downstream tasks for visual recognition.

In this paper, we present a novel formulation for visual pre-training. It originates from the assumption that the best vision model is the one that permits the greatest compression of the observed image data (Barron et al., 1998); in other words, the goal of the pre-training stage is to train the model to better compress image representations. Theoretically, we refer to the Kolmogorov complexity which equals the length of the shortest program to reconstruct image data. However, the Kolmogorov complexity itself is uncomputable, so we instead consider the dual optimization goal in which we train a vision transformer to use a fixed number of tokens to reconstruct the image as accurately as possible.

We implement the above idea using the Semantic Merging and Reconstruction (**SMR**) framework. SMR inherits the encoder-decoder design of MAE (He et al., 2022) where the encoder is used for downstream tasks and the decoder plays an auxiliary role for image reconstruction. Differently, SMR feeds the entire image into the input stage and forces the model to gradually discard less important tokens throughout the encoder. We inject a plug-in module to determine which tokens to discard into each transformer block. The module is easily replaceable and is discarded after the pre-training stage. Hence, at the end of the encoder, the information is stored in a small number of tokens and the decoder must reconstruct the original image with such incomplete information. The reconstruction target can be either raw pixels or the semantic features of the full image extracted by a reference model (*e.g.*, CLIP (Radford et al., 2021)). The latter choice prevents the model from focusing on pixel-level details and leads to better pre-trained models.

Compared to the existing literature of visual pre-training, SMR claims two-fold contributions. **First**, SMR offers a new methodology that explicitly connects visual pre-training with information theory. Although the idea appeared at an early age of AI (*e.g.*, with autoencoders (Hinton & Zemel, 1993; Hinton & Salakhutdinov, 2006)), it has not been well explored in the context of vision transformers. Note that our proxy task is degradation-free, unlike MIM which masks out random patches, or CL which crops image views, which may suffer the degradation dilemma (*i.e.*, a small degradation is insufficient to challenge the target model, yet a large degradation can cause the learning objective irrational). **Second**, SMR applies to various vision transformer architectures, where we showcase the vanilla ViT and two variants in this work. The property that SMR largely reduces the number of tokens in the encoder allows us to balance the pre-training cost and performance (see Table 1) and scales it up to pre-train very large vision models.

| Method | non-deg. | arch-free | mins/ep | mem. | acc. |
|---|---|---|---|---|---|
| MoCo (CL) | | ✓ | $33.2^{\dagger}$ | $20.1G^{\dagger}$ | 83.2% |
| BEiT (MIM) | | | 15.6 | 18.7G | 83.2% |
| MAE (MIM) | | | 9.2 | 16.8G | 83.6% |
| **SMR (ours)** | ✓ | ✓ | 13.7 | 14.1G | **85.4%** |

Table 1: Comparison between SMR and recent approaches for pre-training vision transformers including MoCo (He et al., 2020), BEiT (Bao et al., 2022)), and MAE (He et al., 2022). Here, 'non-deg.' means that the input images are not degraded (*e.g.*, cropped into two views or randomly masked), and 'arch-free' means that the method is easily applicable to any vision transformers. We report the time (mins/ep) and memory (mem.) costs using ViT-B on $8\times$V100 GPUs with a batch size of 128 per GPU ($^{\dagger}$: 64 for MoCo). The last column reports the classification accuracy on ImageNet-1K after 300 epochs of pre-training.

We evaluate SMR using a standard protocol, *i.e.*, pre-training the target model on ImageNet-1K (Russakovsky et al., 2015) and fine-tuning it on ImageNet-1K for classification, on COCO (Lin et al., 2014) for object detection and instance segmentation, and on ADE20K (Zhou et al., 2017) for semantic segmentation. Extensive and competitive results demonstrate the effectiveness of SMR. In particular, after 300 epochs of pre-training, the ViT-B model achieves a top-1 accuracy of 85.4% on ImageNet-1K, a box AP of 53.8% on COCO, and a mIoU of 53.1% on ADE20K, showing the potential of SMR. We also report stronger performance when the backbone is upgraded to HiViT (Zhang et al., 2023) and ConViT (Gao et al., 2022). Additionally, the models pre-trained by SMR enjoy a stronger ability to reconstruct semantics with partial information, revealing a positive correlation between the compactness and effectiveness of visual representations.

## 2 RELATED WORK

The rapid development of computer vision has been driven by deep learning (LeCun et al., 2015), especially deep neural networks. Recently, researchers have realized that vision transformers (Dosovitskiy et al., 2021; Liu et al., 2021) are powerful visual representation learners that outperform convolutional neural networks (Krizhevsky et al., 2012; He et al., 2016; Tan & Le, 2019) in a wide range of computer vision tasks. The plain (vanilla) vision transformers (Dosovitskiy et al., 2021; Chen et al., 2021a; Yuan et al., 2021; Han et al., 2021) assume that the number of tokens remains unchanged. Some variants later emerged called hierarchical vision transformers (Liu et al., 2021; Wang et al., 2021; Zhang et al., 2021; 2023; Tian et al., 2022b), which gradually reduce the feature sizes throughout the backbone. Design decisions were also inherited from the pre-transformer era, such as adding convolutions to vision transformers (Dai et al., 2021; Gao et al., 2022; Srinivas et al., 2021; Gao et al., 2021) and various methods to reduce the computational costs of self-attentions (Liu et al., 2021; Dong et al., 2022; Yang et al., 2021).

Recently, self-supervised learning has attracted widespread attention, which assumes that the weights of vision transformers can be initialized by learning from a set of unlabeled image data. There have been mainly two types of self-supervised learning approaches for vision transformers, namely the contrastive learning (CL) algorithms (Caron et al., 2021; Chen et al., 2021b), which predate vision transformer (Chen et al., 2020; He et al., 2020), and masked image modeling (MIM) algorithms (Bao et al., 2022; He et al., 2022; Xie et al., 2022), which have been reviewed because they better fit the characteristics of vision transformers.

It is still an open problem to understand how self-supervised learning works (Chen et al., 2022) and how it benefits downstream recognition tasks (Zhang et al., 2023; Tian et al., 2022b; Li et al.,

2022b). Nevertheless, researchers have explored many variants beyond the existing methods, including changing the reconstruction target (Wei et al., 2022a;b), degradation types (He et al., 2020; 2022; Tian et al., 2022a), and adapting it to various types of vision transformers (Xie et al., 2022; Gao et al., 2022; Zhang et al., 2023), *etc*.

This paper is also related to the idea of using fewer tokens for visual recognition in the context of vision transformers (Rao et al., 2021; Marin et al., 2021; Kong et al., 2021; Yin et al., 2022; Liang et al., 2022; Li et al., 2022a; Bolya et al., 2023; Long et al., 2022). In these approaches, redundant tokens are omitted or merged throughout the backbone, and redundancy is judged based on whether similar semantic information is represented by other tokens. These methods were developed for image classification, and in this work, we study the application of self-supervised learning by assuming that powerful vision models emerge from compact visual representations.

# 3 METHOD

## 3.1 VISUAL PRE-TRAINING AS IMAGE COMPRESSION

Self-supervised visual pre-training aims to train a model to extract high-quality visual features without relying on semantic labels (*e.g.*, class labels). The key is to find some kind of unsupervised **prior** to constrain the target model. The past years have witnessed an evolution in which the early priors (*e.g.*, understanding the spatial relationship of image patches (Noroozi & Favaro, 2016), filling up missing color (Zhang et al., 2016) or contents (Pathak et al., 2016), *etc*.) have been replaced by two priors that are stronger and more friendly to training vision models. They are (i) the contrastive learning (CL) prior (Chen et al., 2020; He et al., 2020), assuming that the target model shall find the relationship between two random views of an image in a large memory bank, and (ii) the masked image modeling (MIM) prior (Bao et al., 2022; He et al., 2022; Xie et al., 2022), assuming that the target model shall reconstruct the image when many of its patches are masked out.

We note that MIM is related to the information theory, suggesting that a powerful pre-trained model shall have the ability to represent image data using as compressed information as possible. However, MIM achieved the goal in a **passive** manner, *i.e.*, patches are randomly masked out regardless of their semantic importance. This may introduce the so-called degradation dilemma, *i.e.*, the pre-training algorithm needs to set a high masking ratio to challenge the target model, but a high masking ratio increases the risk that important patches are unseen to the target model and thus the reconstruction target becomes unreasonable. The dilemma also holds for contrastive learning where the difficulty and risk lie in the strength of data augmentation (Huo et al., 2021; Wang & Qi, 2022).

To alleviate the dilemma, we advocate for a novel framework for **active** compression. The optimization goal for pre-training remains unchanged, *i.e.*, using fewer tokens to reconstruct the image. But, different from MIM, we feed the entire image to the target model (*i.e.*, we do not mask out patches randomly) and instead ask the model to determine which patches are less important and can be discarded. Without degradation, we can freely increase the compression ratio (*i.e.*, difficulty) without worrying that the reconstruction target may become unreasonable.

The idea of setting data compression as the target for self-supervised learning is not new, *e.g.*, it has been the optimization target for autoencoders (Hinton & Zemel, 1993; Hinton & Salakhutdinov, 2006). However, this objective has not been well studied recently, especially for pre-training vision transformers. We hope to raise the attention of the community to this promising direction.

## 3.2 TOWARDS A HIGH COMPRESSION RATIO OF IMAGE DATA

Let an image dataset be $\mathcal{D} = \{\mathbf{x}_n\}_{n=1}^N$, where $N$ is the number of images and all $\mathbf{x}_n$ are not equipped with semantic labels. The goal is to obtain a deep neural network $f(\mathbf{x}; \boldsymbol{\theta})$ to capture the semantic distribution of these images. In this work, we assume that $f(\mathbf{x}; \boldsymbol{\theta})$ belongs to the family of vision transformers (ViT), where the image is partitioned into patches, embedded into tokens, and the tokens interact with each other through self-attentions into visual representations.

For each image $\mathbf{x}$ (we omit the subscript for simplicity), let the image patches be embedded into a set of tokens, denoted as $\mathcal{T}^{(0)} = \mathrm{PE}(\mathbf{x}) = \{\mathbf{v}_1^{(0)}, \dots, \mathbf{v}_M^{(0)}\}$, where $\mathrm{PE}(\cdot)$ is the patch embedding function, $(0)$ is the block index, and $M$ is the number of tokens. There are a total of $L$ transformer

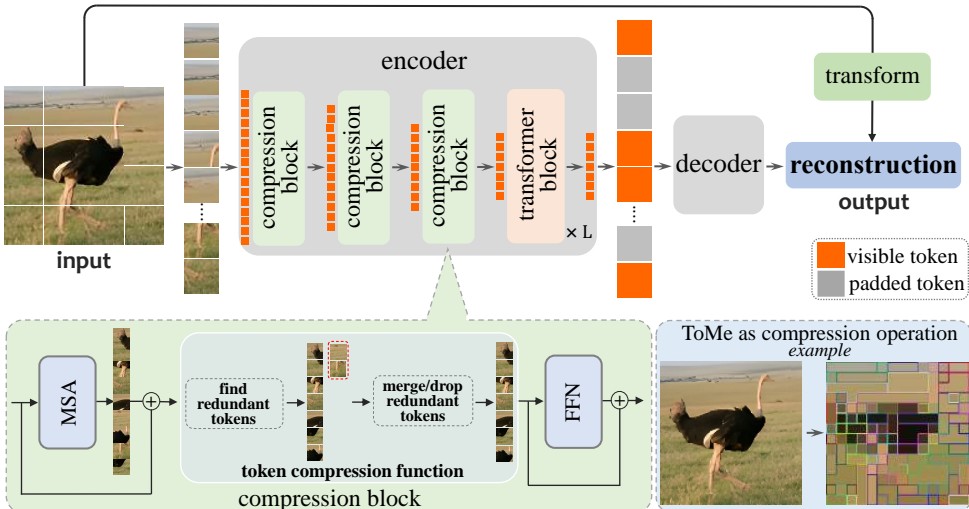

Figure 1: The overall framework of SMR. All tokens are preserved at the input stage. Throughout the encoder stage, a compression function is used to reduce the number of tokens (*e.g.*, by finding less important tokens and dropping/merging them). At the end of the encoder, dummy tokens are filled into the empty slots and a decoder is used for reconstructing raw pixels or CLIP features.

blocks, where the $l$-th block takes $\mathcal{T}^{(l-1)}$ as input and produces $\mathcal{T}^{(l)}$ as output. Let $f^{(l)}(\cdot)$ denote the mathematical function of the $l$-th transformer block that contains some kind of compression operations, then the overall transformer function can be rewritten as

$$f(\mathbf{x}; \boldsymbol{\theta}) \equiv f^{(L)} \circ \ldots \circ f^{(2)} \circ f^{(1)} \circ \mathrm{PE}(\mathbf{x}). \tag{1}$$

Our goal is to optimize $f(\mathbf{x}; \boldsymbol{\theta})$ towards compressing image data. For this purpose, we refer to the Kolmogorov complexity which equals the length of the shortest program that outputs $\mathbf{x}$. However, since the Kolmogorov complexity is uncomputable, we set the learning objective to be using minimal information to reconstruct image data. A simple derivation (see Appendix A) shows that, in the context of vision transformers, the amount of information, $\mathcal{I}(\mathbf{x})$, is proportional to the smallest number of tokens at any layer, *i.e.*, $\mathcal{I}(\mathbf{x}) \propto \min_l M^{(l)}$, where $M^{(l)}$ is the number of tokens at the $l$-th layer. To measure the reconstruction quality of data, we introduce a decoder, $g(\cdot, \boldsymbol{\tau})$, that takes $f(\mathbf{x}; \boldsymbol{\theta})$ as input and produces reconstruction results. Hence, the learning objective is written as:

$$\min_{\boldsymbol{\theta}, \boldsymbol{\tau}} \mathbb{E}_{\mathbf{x} \in \mathcal{D}}[\mathcal{I}(\mathbf{x})], \qquad \text{s.t.} \quad |g(f(\mathbf{x}; \boldsymbol{\theta}), \boldsymbol{\tau}) - h(\mathbf{x})| < \delta. \tag{2}$$

Here, $h(\mathbf{x})$ is a reference model and $\delta$ is a threshold. Setting $h(\mathbf{x}) \equiv \mathbf{x}$ and $\delta = 0$ asks for perfect, pixel-level reconstruction which is often impossible yet meaningless (raw pixels often contain noise and artifacts). So, we allow $\delta > 0$ and set $h(\mathbf{x})$ to be a feature extractor (*e.g.*, CLIP (Radford et al., 2021)) which projects $\mathbf{x}$ to the semantic space (Wei et al., 2022b;c). But, since $\mathcal{I}(\mathbf{x})$ is a discrete variable, optimizing Eqn (2) is intractable. We note that the key of Eqn (2) lies in the tradeoff between information and reconstruction, so we rewrite it into the dual form:

$$\min_{\boldsymbol{\theta}, \boldsymbol{\tau}} \mathbb{E}_{\mathbf{x} \in \mathcal{D}}[|g(f(\mathbf{x}; \boldsymbol{\theta}), \boldsymbol{\tau}) - h(\mathbf{x})|], \qquad \text{s.t.} \quad \min_l M^{(l)} = M', \tag{3}$$

where $M'$ is an integer smaller than $M$. In other words, we ask the target model to reconstruct the original image using a fixed number of tokens. Note that Eqn (3) is friendly to implementation because we can fix the number of tokens at each layer and pack training images into mini-batches.

### 3.3 SEMANTIC MERGING AND RECOVERY

We implement the above formulation using the Semantic Merging and Reconstruction (**SMR**) framework. The design of SMR is illustrated in Figure 1. SMR is degradation-free, *i.e.*, it does not perform random masking in the input stage, but instead asks the target model to recognize and discard redundant (less important) tokens throughout the encoder stage.

Mathematically, we equip each transformer block, say $f^{(l)}(\cdot)$, with a token compression function, $\text{TC}(\cdot; K^{(l)})$, where $K^{(l)}$ denotes the number of tokens to be dropped at this stage, *i.e.*, $\text{TC}(\cdot; K^{(l)})$ reduces $M^{(l)}$ by $K^{(l)}$. The numbers of $\{K^{(1)}, \ldots, K^{(L)}\}$ is called the configuration of compression and $\sum_l K^{(l)} < M$. Let us denote $\hat{f}^{(l)}(\cdot)$ as the variant of $f^{(l)}(\cdot)$ with the token compression function injected. Combining Eqns (1) and (3), the overall pre-training objective is rewritten as

$$\mathcal{L}_{\text{recon}} = \mathbb{E}_{\mathbf{x} \in \mathcal{D}} |g(\hat{f}^{(L)} \circ \ldots \circ \hat{f}^{(1)} \circ \text{PE}(\mathbf{x})) - h(\mathbf{x})|. \tag{4}$$

After the pre-training stage, all $\text{TC}(\cdot; K^{(l)})$ are removed from $\hat{f}^{(l)}(\cdot)$ and the remaining part, $f(\mathbf{x}; \boldsymbol{\theta})$, is used for downstream tasks. Below, we discuss some important design choices.

**Network architecture.** SMR is generalized to various architectures (see Section 4.1). We study three of them in this paper, namely, ViT (Dosovitskiy et al., 2021), HiViT (Zhang et al., 2023), and ConViT (Gao et al., 2022). Since SMR is degradation-free, its application on HiViT and ConViT is straightforward, unlike MIM which needs special and costly treatments to avoid 'information leakage' (Zhang et al., 2023; Gao et al., 2022).

**Reconstruction target.** SMR works well with both the pixel-level and semantic-level reconstruction targets and shows advantage over corresponding baselines (*e.g.*, BEiT (Bao et al., 2022), MAE (He et al., 2022), SD (Wei et al., 2022a), *etc.*). Yet, as we shall see in Section 4.3, using CLIP features as the reconstruction target improves visual pre-training because it is unreasonable to force the target model to reconstruct the raw pixels which may contain noise and artifacts.

**Token compression function.** Here we discuss the function of $\text{TC}(\cdot)$ which aims to compress the number of tokens. Below we investigate some choices as special cases of $\text{CF}(\cdot)$, yet we emphasize that SMR is open to other choices of $\text{CF}(\cdot)$. Each algorithm receives a token set $\mathcal{T}$ and a hyper-parameter $r$ indicating the number of tokens to be compressed, and produces a compressed token set with $r$ tokens fewer and it is possible that part of remain tokens are modified.

- **Token merging (TM).** This algorithm was introduced in ToMe (Bolya et al., 2023) where the token set is randomly partitioned into two equal-sized subsets and each token in the first set is mapped to the most similar token in the second set. Then, the top-$r$ tokens in the first set with highest redundancy are averaged to the corresponding token in the second set.

- **Token dropping (TD).** We use the same strategy as TM to choose the top-$r$ most redundant tokens, but we directly remove them instead of averaging them into other tokens.

- **Inattentive token fusion (ITF).** This algorithm was introduced in EViT (Liang et al., 2022) where each token is assigned with an attentiveness score computed by self-attention. Then, the top-$r$ inattentive ones are weighted averaged into one token and the remaining (attentive) tokens are preserved.

- **Random dropping (RD).** We randomly remove $r$ tokens from the token set.

As shown in Section 4.3, stronger compression functions are helpful in preserving useful semantic information and assisting learning representations.

## 3.4 DISCUSSIONS

**MIM as a special case of SMR.** MIM can be formulated using the above framework. If we perform token dropping before the first transformer block (SMR does not allow it), *i.e.*, setting $K^{(0)}$ to be a specific value (*e.g.*, $3/4$ of the total number of tokens) and all other $K^{(l)} = 0$, the framework degenerates to MIM. However, since all tokens before the first transformer block are independent, the selection of dropped tokens is random and may affect semantic understanding. As we shall see in Appendix E, SMR learns to preserve important tokens for semantic reconstruction.

**Advantages.** We analyze the difference between SMR and MIM from two aspects. First, SMR does not perform random masking on the input image, so it alleviates the burden of setting an irrational reconstruction target via masking out essential content from the input image. To the best of our knowledge, SMR is the first pre-training proxy for vision transformers which allows for changing the target difficulty but does not perform input degradation. Second, SMR learns to actively select and remove redundant tokens, so it gains the ability to represent visual semantics using less

information: this aligns with the working mechanism of autoencoders (Kingma & Welling, 2013) that learn stronger representations by improving the compactness of visual features.

**Computational complexity.** Since the reconstruction is measured in the semantic space, the baseline method uses the entire image (with part of patches replaced with [MASKED] tokens, as in BEiT (Bao et al., 2022) and SD (Wei et al., 2022c)) as input, meanwhile the number of tokens remains unchanged throughout the encoder. In comparison, SMR enjoys a lower computational complexity because the token number is gradually reduced. Let a vanilla vision transformer have $L$ layers and the $l$-th layer has $M^{(l)}$ tokens, the time and memory costs of pre-training are proportional to $\mathcal{O}(\sum_l M^{(l)2})$ and $\mathcal{O}(\sum_l M^{(l)})$, respectively. BEiT and SD keep $M^{(l)} \equiv M$ for all $l$ and thus the numbers are $\mathcal{O}(LM^2)$ and $\mathcal{O}(LM)$. Instead, SMR reduces the number of tokens at an early stage (see diagnostics in Section 4.3 which show that early token compression leads to better results) and thus a large portion layers have a smaller $M^{(l)}$. In the pre-training stage, SMR is 15%–20% faster than BEiT and SD: the advantage is made smaller by the decoder stage and the teacher in which all methods have the same complexity and the advantage would be amplified with the pre-training of larger models.

On the other hand, the advantage in memory usage is even larger, which allows us to pre-train very large vision models. In practice, we scale up a ViT model into 50 layers with the dimension being 4,096. This model has more than 10B parameters and is unable to fit into GPUs with 80G memory (*e.g.*, NVIDIA Tesla-A800) using MIM or SD. Thanks to SMR, the memory usage is reduced and, for the first time, we can train the 10B model without model partitioning (details in Appendix F).

# 4 EXPERIMENTS

## 4.1 QUANTITATIVE STUDIES

We evaluate SMR following the standard protocol, *i.e.*, pre-training the model on ImageNet-1K (Russakovsky et al., 2015) (without using semantic labels) followed by fine-tuning it on ImageNet-1K, COCO (Lin et al., 2014), and ADE20K (Zhou et al., 2017). All implementation details are provided in Appendix B.

**ImageNet classification.** The classification accuracy on ImageNet-1K is summarized in Table 2. One can see that SMR outperforms the competitors on all three transformer architectures. Specifically, on the vanilla ViT-B, the advantages of SMR over the three direct baselines (*i.e.*, MAE for MIM, FD-CLIP for SD, and BEiT-v2 for their combination) are 1.8%, 0.5%, and 0.3%, respectively. The good practice also transfers to the ViT-L backbone. By using CLIP-L as the teacher model, SMR reports an 87.7% classification accuracy after 300 epochs of pre-training and 50 epochs of fine-tuning on ImageNet-1K.

**COCO and ADE20K.** Results of the downstream recognition tasks are sum-

| Method | Arch. | Sup. | Eps. | Param. (M) | FT acc. |
|---|---|---|---|---|---|
| MoCo v3 (Chen et al., 2021b) | ViT-B | pixel | 300 | 86 | 83.2 |
| iBoT (Zhou et al., 2021) | ViT-B | pixel | 1600 | 86 | 84.0 |
| BEiT (Bao et al., 2022) | ViT-B | DALL-E | 400 | 86 | 83.2 |
| SimMIM (Xie et al., 2022) | Swin-B | pixel | 800 | 88 | 84.0 |
| MaskFeat (Wei et al., 2022a) | ViT-B | HOG | 800 | 86 | 84.0 |
| CAE (Chen et al., 2022) | ViT-B | DALL-E | 800 | 86 | 83.6 |
| MVP (Wei et al., 2022b) | ViT-B | CLIP-B | 300 | 86 | 84.4 |
| CAE-v2 (Zhang et al., 2022) | ViT-B | CLIP-B | 300 | 86 | 85.3 |
| MAE (He et al., 2022) | ViT-B | pixel | 1600 | 86 | 83.6 |
| FD-CLIP (Wei et al., 2022c) | ViT-B | CLIP-B | 300 | 86 | 84.9 |
| BEiT-v2 (Peng et al., 2022) | ViT-B | CLIP-B | 300 | 86 | 85.0 |
| **SMR (ours)** | ViT-B | CLIP-B | 300 | 86 | **85.4** |
| **SMR (ours)** | ViT-B | CLIP-B | 800 | 86 | **85.7** |
| **SMR (ours)** | ViT-B | CLIP-L | 300 | 86 | **86.7** |
| HiViT (Zhang et al., 2023) | HiViT-B | pixel | 300 | 79 | 84.6 |
| FD-CLIP[†] (Wei et al., 2022c) | HiViT-B | CLIP-B | 300 | 79 | 85.4 |
| BEiT-v2[†] (Peng et al., 2022) | HiViT-B | CLIP-B | 300 | 79 | 85.4 |
| **SMR (ours)** | HiViT-B | CLIP-B | 300 | 79 | **85.7** |
| ConvMAE (Gao et al., 2022) | ConViT-B | pixel | 300 | 88 | 85.0 |
| FD-CLIP[†] (Wei et al., 2022c) | ConViT-B | CLIP-B | 300 | 88 | 85.5 |
| BEiT-v2[†] (Peng et al., 2022) | ConViT-B | CLIP-B | 300 | 88 | 85.4 |
| **SMR (ours)** | ConViT-B | CLIP-B | 300 | 88 | **85.7** |

Table 2: Top-1 classification accuracy (%) by fine-tuning (FT) the pre-trained models on ImageNet-1K. We compare models of different backbones including ViT (Dosovitskiy et al., 2021), HiViT (Zhang et al., 2023), and ConViT (Gao et al., 2022). [†]: these numbers are reported by our re-implementation.

marized in Table 3. SMR persists the advantages observed in the image classification experiments. On COCO, SMR reports better box/mask APs when either head (Mask R-CNN or Cascade Mask R-CNN) is used. Again, it is worth noting that SMR is easily transplanted to different network architectures and achieves higher downstream recognition accuracy using stronger backbones.

| Method | Arch. | Sup. | Eps. | Param. (M) | COCO MR, 1× | COCO CMR, 3× | ADE20K UPerHead |
|---|---|---|---|---|---|---|---|
| MoCo-v3 (Chen et al., 2021b) | ViT-B | pixel | 300 | 86 | 45.5/40.5 | – | 47.3 |
| BEiT (Bao et al., 2022) | ViT-B | DALL-E | 400 | 86 | 42.1/37.8 | – | 47.1 |
| iBoT (Zhou et al., 2021) | ViT-B | pixel | 1600 | 86 | – | 51.2/44.2 | 50.0 |
| MAE (He et al., 2022) | ViT-B | pixel | 1600 | 86 | 48.4/42.6 | – | 48.1 |
| SimMIM (Xie et al., 2022) | Swin-B | pixel | 800 | 88 | – | – | 52.8 |
| CAE (Chen et al., 2022) | ViT-B | DALL-E | 1600 | 86 | 50.0/44.0 | – | 50.2 |
| MVP (Wei et al., 2022b) | ViT-B | CLIP-B | 300 | 86 | – | 53.5/46.3 | 52.4 |
| FD-CLIP (Wei et al., 2022c) | ViT-B | CLIP-B | 300 | 86 | – | – | 52.8 |
| HiViT (Zhang et al., 2023) | HiViT-B | pixel | 1600 | 66 | 49.5/43.8 | – | 51.2 |
| ConvMAE (Gao et al., 2022) | ConViT-B | pixel | 1600 | 88 | – | – | 51.7 |
| BEiT-v2 (Peng et al., 2022) | ViT-B | CLIP-B | 300 | 86 | – | – | 52.7 |
| **SMR (ours)** | ViT-B | CLIP-B | 300 | 86 | **52.0/45.0** | **53.8/46.5** | **53.1** |
| **SMR (ours)** | HiViT-B | CLIP-B | 300 | 79 | **52.6/45.4** | **54.3/47.0** | **54.0** |
| **SMR (ours)** | ConViT-B | CLIP-B | 300 | 88 | **52.6/45.5** | **54.4/47.1** | **54.2** |

Table 3: Downstream recognition results (%) on COCO (for object detection and instance segmentation) and ADE20K (for semantic segmentation). We report box/mask AP on COCO and mIoU on ADE20K. Abbreviations: MR is for Mask R-CNN and CMR is for Cascade Mask R-CNN.

## 4.2 QUALITATIVE STUDIES OF IMAGE COMPRESSION

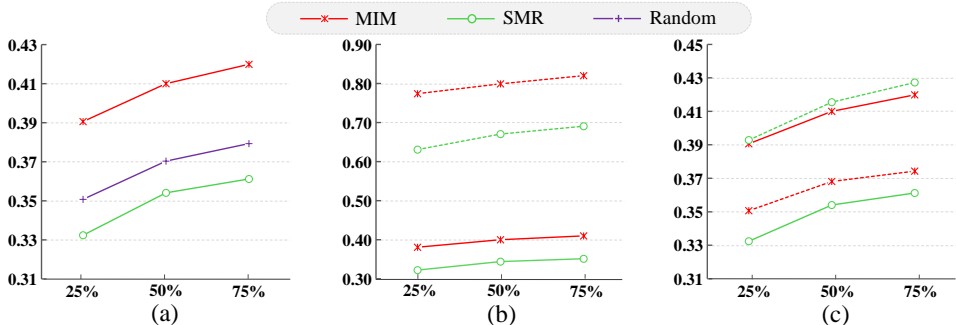

Figure 2: Average reconstruction loss with respect to the proportion of masked/compressed tokens. **(a)** Each pre-trained model is directly tested on the task that it was trained with. The red and black lines are copied to (b) and (c) for comparison. **(b)** The MIM-trained model is tested on the SMR task (the red dashed line); the SMR-trained model is tested on the MIM task (the green dashed line). **(c)** The MIM-trained model is tested on the SMR task after being fine-tuned shortly with SMR (the red dashed line); the SMR-trained model is also fine-tuned with MIM (the green dashed line).

**SMR pre-training improves the compactness of visual representations.** To better compare the pre-training results of MIM (enhanced by SD (Wei et al., 2022c)) and SMR, we evaluate the pre-trained ViT-B models on two reconstruction tasks, *i.e.*, the MIM task where a random subset of tokens are masked from input and the SMR task where a subset of tokens are actively chosen and dropped throughout the encoder. The total numbers of dropped tokens for MIM and SMR are the same. We set three compression ratios (25%, 50%, 75%) and record the average reconstruction loss over a fixed subset of the ImageNet-1K validation set. Figure 2 summarizes the results, where the $x$-axis shows the ratio of compressed tokens and the $y$-axis shows the reconstruction loss.

Clearly, there is a tradeoff between the feature compactness (positively correlated to the compression ratio) and the quality of reconstruction. We find that SMR achieves a better tradeoff than MIM, either when the model is directly tested on the same task or transferred to the other task. In other words, SMR acquires a better ability to use few tokens to represent the image semantics. As we shall see in the subsequent experiments, SMR reports better downstream recognition results than MIM and SD. Hence, SMR validates the intrinsic connection between the compactness and effectiveness of visual representations, and offers an efficient implementation for vision transformers.

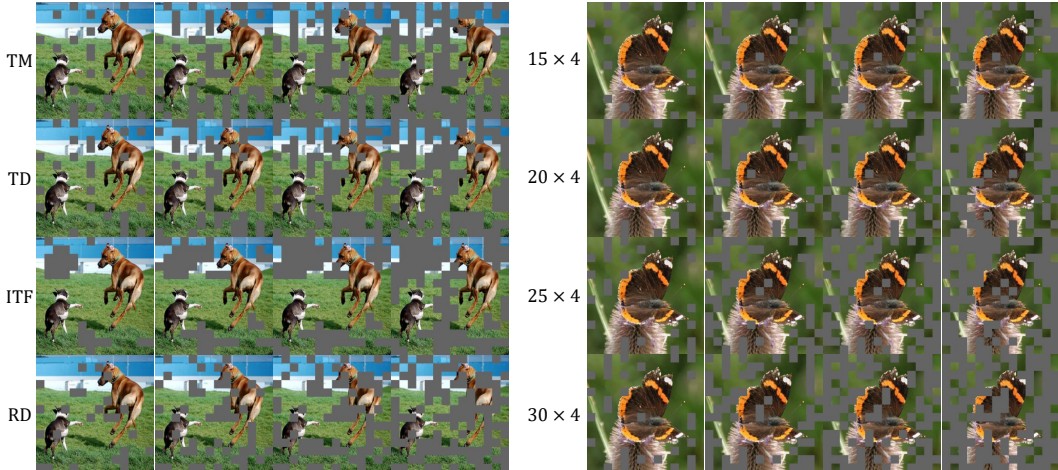

Figure 3: Visualization of token compression and image reconstruction. We compare different token compression functions and different configurations. The default configuration in the left part is $25 \times 4$ and the default compression function in the right part is TM.

Lastly, in Figure 3, we visualize the impact of different token compression functions and compression ratios. All three active functions (ITF, TM, TD) report similar results, and TM reports the smallest reconstruction loss. Additionally, a moderate ratio (*e.g.*, $25 \times 4$) leads to the best trade-off between compression and reconstruction. These nice properties correspond to better pre-trained models, see Tables 4f and 4e. More examples of visualization are available in Appendix E.

## 4.3 DIAGNOSTIC STUDIES

Table 4 summarizes the diagnostic results on ImageNet-1K classification. Unless specified, we use 100 epochs of pre-training followed by 100 epochs of fine-tuning. These experiments allow us to better understand the properties of the proposed SMR framework. We choose token merging (ToMe) as the compression operation by default.

**Reconstruction target: Table 4a.** We investigate two reconstruction targets, the raw pixels and the CLIP features. Interestingly, SMR with CLIP features outperforms both the SD and MIM+SD methods, but SMR with raw pixels reports inferior results to the MIM baseline (He et al., 2022). This

| Method | Sup. | Acc. |
|---|---|---|
| SD ($SMR_0$) | pixel | 81.8 |
| SD ($SMR_0$) | CLIP-B | 84.8 |
| $SMR_{25 \times 4}$ | pixel | 82.2 |
| $SMR_{25 \times 4}$ | CLIP-B | 85.1 |

(a) Reconstruction Target

| Method | Blocks | Acc. |
|---|---|---|
| $SMR_{25 \times 4}$ | 1 | 84.9 |
| $SMR_{25 \times 4}$ | 2 | 85.1 |
| $SMR_{25 \times 4}$ | 4 | 83.2 |
| $SMR_{25 \times 4}$ | 6 | 82.1 |

(b) Length of Decoder

| Method | Eps. | Acc. |
|---|---|---|
| SD ($SMR_0$) | 100 | 84.8 |
| SD ($SMR_0$) | 300 | 85.0 |
| $SMR_{25 \times 4}$ | 100 | 85.1 |
| $SMR_{25 \times 4}$ | 300 | 85.4 |

(c) Difficulty

| Method | Drop Alg. | Acc. |
|---|---|---|
| $SMR_{15 \times 4}$ | RD | 84.2 |
| $SMR_{25 \times 4}$ | RD | 84.4 |
| $SMR_{15 \times 4}$ | TM | 84.7 |
| $SMR_{25 \times 4}$ | TM | 85.1 |

(d) Dropping Strategy

| Method | Drop Alg. | Acc. |
|---|---|---|
| $SMR_0$ | – | 84.8 |
| $SMR_{25 \times 4}$ | TM | 85.4 |
| $SMR_{25 \times 4}$ | TD | 85.2 |
| $SMR_{25 \times 4}$ | ITF | 85.0 |
| $SMR_{25 \times 4}$ | RD | 84.4 |

(e) Compress. Function

| Method | # Drop | Acc. |
|---|---|---|
| $SMR_{15 \times 4}$ | 60 | 84.7 |
| $SMR_{20 \times 4}$ | 80 | 85.0 |
| $SMR_{25 \times 4}$ | 100 | 85.1 |
| $SMR_{30 \times 4}$ | 120 | 85.0 |
| $SMR_{35 \times 4}$ | 140 | 84.8 |

(f) # Dropped Tokens

| Method | # Drop | Acc. |
|---|---|---|
| $SMR_{10 \times 10}$ | 100 | 84.6 |
| $SMR_{20 \times 5}$ | 100 | 84.9 |
| $SMR_{25 \times 4}$ | 100 | 85.1 |
| $SMR_{50 \times 2}$ | 100 | 84.6 |
| $SMR_{100 \times 1}$ | 100 | 84.3 |

(g) Pace of Dropping

| Method | B-IDs | Acc. |
|---|---|---|
| $SMR_{25 \times 4}$ | 1–4 | 85.1 |
| $SMR_{25 \times 4}$ | 3–6 | 84.2 |
| $SMR_{25 \times 4}$ | 5–8 | 84.5 |
| $SMR_{25 \times 4}$ | 7–10 | 84.4 |
| $SMR_{25 \times 4}$ | 9–12 | 84.4 |

(h) Position of Dropping

Table 4: Diagnostic studies. $SMR_{K' \times L'}$ means that $K'$ tokens are compressed after each of the $L'$ transformer blocks. To save space, we only show the factor(s) to be diagnosed in each subtable. The default parameters are: using CLIP-B as supervision, the decoder length is 2, 100 pre-training epochs, using TM for token compression, and token compression happens in the first $L'$ blocks.

is explained by the nature of SMR: the goal is to represent the image contents with fewer tokens, so compared to recovering raw pixels that may contain random artifacts or noise, a better solution is to recover high-level semantic features.

**Length of decoder: Table 4b.** Unlike in MIM, using a heavy decoder (*e.g.*, with 6 or 4 transformer blocks) significantly harms the pre-training performance of SMR. This is because we have used CLIP features as the reconstruction target, and we hope that the features at the end of the encoder have a close relationship to the CLIP features.

**Difficulty and comparison to SD: Table 4c.** Token compression makes the pre-training task of SMR more difficult, allowing the model to benefit from longer training epochs, *e.g.*, from using 100 to 300 pre-training epochs, the improvement is $0.2\%$ for SD and $0.3\%$ for SMR. Additionally, when the pre-training is extended to $800$ epochs, SD suffers an $0.3\%$ accuracy drop arguably because the model over-fits the CLIP features, yet SMR is almost unaffected. However, as analyzed in Table 4f, setting an over-high difficulty can still deteriorate the pre-trained model.

**Choice of token compression function, Tables 4d and 4e.** These experiments are pre-trained for 300 epochs. Among the different token compression functions, the token merging operation (TM) achieves the best results. Interestingly, the random dropping operation reports even worse accuracy than the baseline, while the other three functions surpass the baseline. This indicates that semantic-aware compression functions help to improve the compactness of visual representations. We also compare TM and RD with different numbers of dropped tokens. TM consistently outperforms RD, and also obtains a larger gain when the dropping ratio increases (from $15 \times 4$ to $25 \times 4$). Combining with the observation in Figure 2 that TM reports a smaller reconstruction loss, we conclude that dropping semantically redundant tokens is a better option for compression-based visual pre-training.

**Number of compressed tokens, Table 4f.** Setting a moderate number of compressed tokens (*i.e.*, a moderate difficulty) leads to the highest accuracy. Although SMR allows for a higher pre-training difficulty, the best strategy does not lie in removing more tokens (*i.e.*, reducing spatial information). Other possibilities include reducing the feature dimensionality, which we will study in the future.

**Pace of token compression, Table 4g.** Setting a moderate pace leads to the highest accuracy, because an over-fast pace can force the model to drop some critical tokens, yet an over-slow pace can postpone the bottleneck which, according to the next paragraph, brings a negative impact.

**Position of token compression, Table 4h.** It is interesting that compressing tokens at the very first blocks produces the highest accuracy. We conjecture that the 'bottleneck' (*i.e.*, where token-compression ends) shall appear early to leave the remaining part of the encoder to learn semantic reconstruction. This aligns with MIM where masking happens at the input layer.

**Summarizing the above analysis** on the configuration of token compression, we find two pairs of tradeoffs: one is between pursuing the difficulty of pre-training and ensuring the quality of reconstruction, and the other is between a fast compression pace (for an earlier bottleneck) and a low compression loss. We leave them as open problems for future research.

## 5 CONCLUSIONS

In this paper, we propose Semantic Merging and Reconstruction (**SMR**) as a novel approach for pre-training vision transformers. The methodology originates from the information theory. In practice, we force the target model to drop image tokens throughout the encoder and the decoder still has the ability of image reconstruction. We discuss several design choices including the reconstruction target, token compression function, and dropping options. SMR shows competitive performance in a few downstream visual recognition tasks.

An intriguing takeaway of this paper lies in the insight that vision transformers benefit from the objective of learning compact visual representations. Specifically, SMR offers a non-degradation framework that paves the way to continue increasing the difficulty of pre-training tasks (*e.g.*, by increasing the compression ratio) without worrying about the rationality of difficult reconstruction tasks. We look forward to future research efforts in this direction. Open topics include (1) applying stronger algorithms for token compression, (2) exploring the possibility of information reduction from other dimensions (*e.g.*, compressing the number of channels for each token), (3) designing specialized network architectures towards a higher ratio of token compression, *etc*.

ETHICS STATEMENT

This paper focuses on visual pre-training and thus has no additional ethical concerns beyond a large corpus of research in this field. We do not introduce any new data and only made use of existing image datasets (*e.g.*, ImageNet, COCO, ADE20K, *etc.*) and pre-trained models (*e.g.*, CLIP).

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

## A   THE CONNECTION BETWEEN MINIMAL INFORMATION AND THE NUMBER OF TOKENS

Here we explain why the amount of information to reconstruct $\mathbf{x}$ (*i.e.*, $\mathcal{I}(\mathbf{x})$) is proportional to the smallest number of tokens at any layer. Please refer to Section 3 for the background and mathematical notations.

According to the mechanism of vision transformers, (approximately) reconstructing $\mathbf{x}$ can be done by taking out the features at any layer and forward propagating them through the subsequent part of the model. That said, $\mathcal{I}(\mathbf{x})$ is composed of two parts, (i) the features and (ii) a part of the vision transformer weights and configurations. Part (i) is specific for each image and Part (ii) is shared among all images.

Note that visual pre-training tends to minimize $\mathcal{I}(\mathbf{x})$ on a very large dataset that potentially contains all possible images in the world. When the number of images goes to infinity, Part (ii) is sufficiently amortized so that its value is negligible, and Part (i) is the only information that we care about. In our setting, the configuration of the target model is known, and so is the number of tokens at each layer (after token compression). Therefore, the most efficient way for image reconstruction is to choose any layer with the smallest number of tokens and use the set of tokens (and the subsequent part of the pre-trained model) for reconstruction. Given a constant feature dimensionality, we conclude that $\mathcal{I}(\mathbf{x}) \propto \min_l M^{(l)}$.

The above derivation assumes that the architecture and configuration remain unchanged, but actually, these factors can also be optimized. In this paper, we simply inherit classical architectures and apply heuristic search to find better configurations. We leave a more efficient search paradigm to future work.

## B   IMPLEMENTATION DETAILS

In this section, we provide the implementation details to pre-train and fine-tune the models.

**ImageNet pre-training.**   We pre-train the base-level model using three network architectures, namely, the vanilla ViT (Dosovitskiy et al., 2021), HiViT (Zhang et al., 2023), and ConViT (Gao et al., 2022). The pre-training stage elapses 300 epochs (we also pre-train ViT-B for 800 epochs) where the first 20 epochs are used for warm-up. We use an AdamW optimizer (Loshchilov & Hutter, 2017) with an initial learning rate of $1.5 \times 10^{-3}$ and it anneals following a cosine schedule. The weight decay is set to be $0.05$ and batch size set to be $2,048$. For each sampled training image, we perform a standard rescaling augmentation, randomly crop a $224 \times 224$ sub-image, and partition it into $14 \times 14$ patches (each patch has $16 \times 16$ pixels). We will provide more details in the Table 5 and Table 6.

**ImageNet classification.**   We append a linear layer to the end of the pre-trained encoder and fine-tune the weights of the entire model. We use the AdamW optimizer with a total of 100 fine-tuning epochs and 5 warm-up epochs. An initial learning rate of $5 \times 10^{-4}$ with layer decay of $0.65$ is used and it anneals following a cosine schedule. The weight decay is set to be $0.05$ and the batch size is set to be $1,024$.

**COCO detection and segmentation.**   The settings mostly follow CAE (Chen et al., 2022). We use two detection heads, Mask R-CNN (He et al., 2017) with a $1\times$ training schedule (12 epochs) and Cascade Mask R-CNN (Cai & Vasconcelos, 2019) with a $3\times$ training schedule (36 epochs), both of which are implemented by the MMDetection (Chen et al., 2019) library. We use the AdamW optimizer (Loshchilov & Hutter, 2017) with a weight decay of $0.05$. The initial learning rate is $3 \times 10^{-4}$ and it decays twice by a factor of 10, after $3/4$ and $11/12$ of fine-tuning epochs, respectively. The layer-wise decay rate is set to be $0.75$, $0.85$, and $0.75$ for ViT, HiViT, and ConViT, respectively. We apply multi-scale training and single-scale testing.

**ADE20K segmentation.**   We follow BEiT (Bao et al., 2022) to append a UPerHead (Xiao et al., 2018) to the end of the backbone. We use the AdamW optimizer (Loshchilov & Hutter, 2017) and the learning rate is set to $3 \times 10^{-5}$. We fine-tune the model for a total of 160K iterations and the batch size is set to be 16. The input resolution is set to be the default value, $512 \times 512$, and we do not perform multi-scale testing.

We provide more implementation details here. The pre-training and fine-tuning details are provided in Tables 5 and 6, respectively. During the pre-training stage, we do **not** use other training techniques like the early-stage supervision at $3/4$ of the third stage like BEiT-v2 (Peng et al., 2022).

## C   MORE ABLATIVE STUDIES

We provide more ablative studies in Table 7. All the results are obtained by pre-training the ViT-B model for 100 epochs and fine-tuning it for 100 epochs. These results are complementary to the results in Table 4 of the main article.

These additional experiments are mainly about the compression configuration of SMR. Specifically,

- In Table 7a, we conduct different experiments about compression frequency for Blocks 1–4 and we find that an even compression strategy leads to the best accuracy.
- In Table 7b, we provide the results of different compression paces of SMR, and all the obtained results are slightly worse than the default configurations.
- We report more results about the number of compressed tokens in Table 7c, which implies that a moderate difficulty leads to the best performance of pre-training.

## D   COMPUTATIONAL EFFICIENCY

We compare the pre-training complexity of SMR and SD in Table 8. One can see that SMR requires fewer computational costs during the pre-training stage, especially for ViT and HiViT. Note that ConViT contains more FLOPs in the first 2 stages, which are not saved by SMR, so we observe a smaller ratio of saved computational costs in ConViT.

Table 5: Hyperparameters for pre-training on ImagetNet-1K.

| Hyperparameters | ViT-B | HiViT-B | ConvViT-B |
|---|---|---|---|
| Patch size | | 16 | |
| Hidden size | 768 | 512 | 768 |
| Layers | 12 | 3-3-24 | 2-2-11 |
| FFN hidden size | 3072 | 2048 | 3072 |
| Attention heads | 12 | 8 | 12 |
| Attention head size | | 64 | |
| Params. (M) | 86 | 79 | 88 |
| Hierarchical ViT | ✗ | ✓ | ✓ |
| Input resolution | | $224\times224$ | |
| Training epochs | | 300 | |
| Warmup epochs | | 30 | |
| Batch size | | 2048 | |
| Optimizer | | AdamW | |
| Peak learning rate | | 1.5e-3 | |
| Mininal learning rate | | 2e-5 | |
| Learning rate schedule | | cosine decay | |
| Gradient clipping | | None | |
| Weight decay | | 0.05 | |
| Optimizer momentum | | $\beta_1,\beta_2 = 0.9,0.98$ | |
| Optimizer $\epsilon$ | | 1e-8 | |
| Stoch. depth | | 0.1 | |
| Augmentation | | RandomResizeCrop | |
| APE | | ✓ | |
| RPE | | ✗ | |

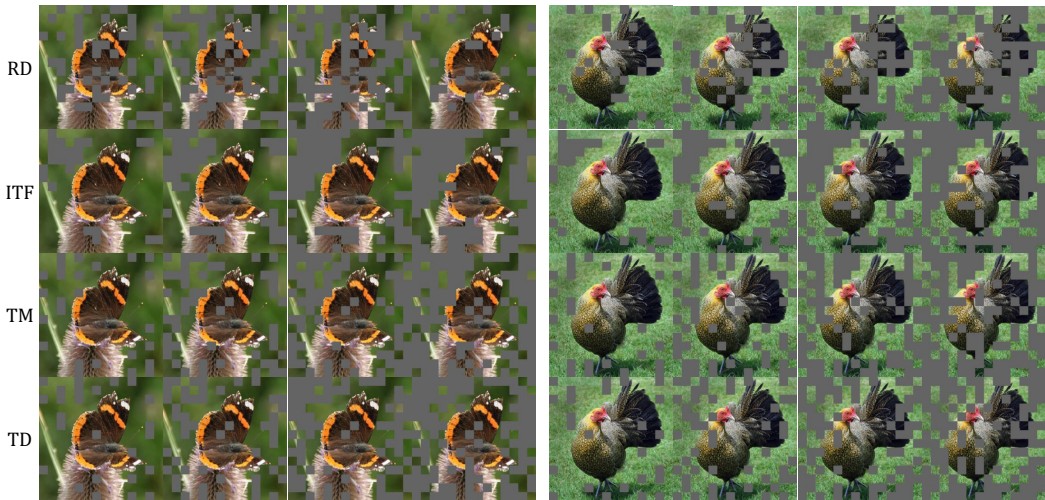

Figure 4: Visualization of token compression and image reconstruction. We compare different token compression functions with the default setting, *i.e.*, dropping 25 tokens in each of the first 4 blocks.

# E  VISUALIZATION

We visualize more examples of token merging in Figure 5, including some images from the COCO and ADE20K datasets. We show more examples of compression functions in Figure 4.

Table 6: Hyperparameters for fine-tuning on ImagetNet-1K.

| Hyperparameters | ViT-B | HiViT-B | ConvViT-B |
|---|---|---|---|
| Input resolution | | $224 \times 224$ | |
| Training epochs | | 100 | |
| Batch size | | 1024 | |
| Warmup epochs | | 20 | |
| Optimizer | | AdamW | |
| Peak learning rate | | 5e-4 | |
| Mininal learning rate | | 2e-6 | |
| Learning rate schedule | | cosine decay | |
| Weight decay | | 0.05 | |
| Layer decay | 0.65 | 0.85 | 0.75 |
| Optimizer momentum | | $\beta_1, \beta_2 = 0.9, 0.999$ | |
| Optimizer $\epsilon$ | | 1e-8 | |
| Label smoothing | | 0.1 | |
| Stoch. path | | 0.2 | |
| Dropout | | ✗ | |
| Gradient clipping | | None | |
| Repeated Aug. | | None | |
| Augmentation | | RandAug (9,0.5) | |
| Erasing prob. | | 0.25 | |
| Mixup prob. | | 0.8 | |
| Cutmix prob. | | 1.0 | |
| Color jitter | | 0.4 | |
| APE | | ✓ | |
| RPE | | ✓ | |

## F  SCALING TO 10B MODEL

Thanks to SMR that reduces the time and memory costs of pre-training, we can pre-train a very large vision model at an affordable overhead. In particular, we design a 10B model (with more than 10 billion parameters). It is a vanilla vision transformer with 50 blocks and each token has a feature dimension of 4,096. We use the DeepSpeed library to save memory. Note that, even with DeepSpeed, the existing pre-training methods including MIM (Bao et al., 2022) and SD (Wei et al., 2022c) run out of memory even on NVIDIA Tesla-A800 GPUs (with 80GB memory).

We pre-train the 10B model on ImageNet-21K for 10 epochs using $8\times$ Tesla-A800 GPUs and fine-tune the model on ImageNet-1K using the $224 \times 224$ resolution for 10 epochs. The model reports an $86.2\%$ classification accuracy, leaving space for future improvement. Our experiment showcases the potential of SMR in pre-training very large vision models with limited computational budgets.

| Block | # Compress | Acc. |
|---|---|---|
| 1-2-3-4 | 25-25-25-25 | 85.1 |
| 1-2-3-4 | 40-30-20-10 | 84.8 |
| 1-2-3-4 | 10-20-30-40 | 84.9 |
| 1-2-3-4 | 50-0-0-50 | 84.8 |
| 1-2-3-4 | 50-0-50-0 | 84.8 |
| 1-2-3-4 | 0-50-0-50 | 84.7 |

(a) Position of Dropping

| Block | # Compress | Acc. |
|---|---|---|
| 1-3-5-7 | 25-25-25-25 | 85.0 |
| 1-3-5-7 | 40-30-20-10 | 84.8 |
| 1-3-5-7 | 10-20-30-40 | 84.9 |
| 1-4-7-11 | 25-25-25-25 | 84.9 |
| 1-4-7-11 | 40-30-20-10 | 84.7 |
| 1-4-7-11 | 10-20-30-40 | 84.9 |

(b) Pace of Dropping

| Method | B-IDs | Acc. |
|---|---|---|
| $25 \times 4$ | 1–4 | 85.1 |
| $25 \times 5$ | 1–5 | 85.0 |
| $25 \times 6$ | 1–6 | 84.8 |
| $30 \times 5$ | 1–5 | 84.9 |
| $20 \times 5$ | 1–5 | 84.9 |

(c) # Dropped Tokens

Table 7: More diagnostic studies. Here, 'Block' means the position of token compression and 'Compress' is the corresponding number of compressed tokens.

Table 8: Pre-training complexity (FLOPs) comparison under the default configuration. We do not count the complexity of the teacher model.

| Method | ViT-B (Dosovitskiy et al., 2021) | HiViT (Zhang et al., 2023) | ConvViT (Gao et al., 2022) |
|---|---|---|---|
| baseline | 17.6 | 18.1 | 23.2 |
| SMR | 12.8 | 13.3 | 19.1 |

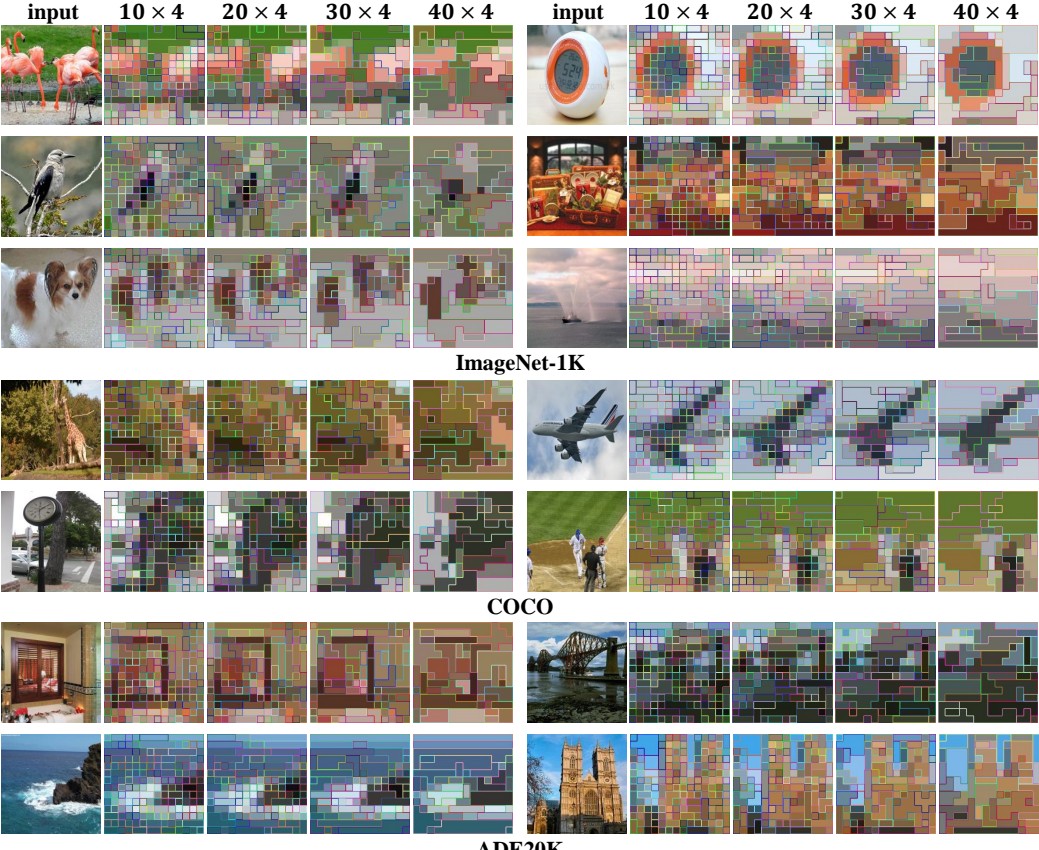

Figure 5: More examples of token dropping as compression operation. For each image, we study the cases where the $10 \times 4$, $20 \times 4$, $30 \times 4$, and $40 \times 4$ of $14 \times 14 = 196$ tokens are compressed after the first four transformer blocks. We visualize the preserved tokens. We provide image examples picked from ImageNet-1K, COCO, and ADE20K.

