# OpenReview forum: "Image Compression Is an Effective Objective for Visual Representation Learning"
_ICLR.cc/2024/Conference — ICLR 2024 Conference Withdrawn Submission_

### Official Review · Reviewer_TTbD · 2023-10-30

**Soundness:** 2 fair
**Presentation:** 3 good
**Contribution:** 2 fair
**Rating:** 5
**Confidence:** 5

**Summary:**

This paper proposed a new visual representation learning method, named Semantic Merging and Reconstruction (SMR). SMR follows the principles of compressing images. It forces the encoder to drop image tokens and reconstructs the image representation from the decoder. Experiments on ImageNet, COCO and ADE20k have shown its effectiveness.

**Strengths:**

1. This paper proposes a new visual self-supervised training framework. Different from MIM method, it lets the encoder to actively drop less important tokens, thereby compressing the image while preserving important information.

2. While the model makes use of pretrained CLIP models, experiments on downstream tasks show good performances.

**Weaknesses:**

1. Unclear deduction from information theory. My major concern is the deduction from information theory, which is also claimed to be the first contribution by the paper.

* First, from the perspective of information theory, the learning objective is, as the paper writes, “using minimal information to reconstruct image data”. The amount of information is proportional to the smallest number of tokens in the case of ViT. This implies that we need to get the model output as few tokens as possible. But Equation (2) is just optimizing the model parameters, then how is information minimization achieved? The optimization variables should be related to token number, but the relationship is unclear.

* Another problem is how the dual form Equation (3) is obtained from Equation (2)? The transition does not seem straight-forward. The authors are suggested to provide derivation.

2. Comparison with previous work[1]. A previous work[1], MILAN, has proposed to use semantic aware sampling to guide image masking. It avoids the random selection problem in MIM, and also achieves high performance by using CLIP as teacher (For ViT-B, it obtains 85.4% accuracy on ImageNet, 52.6 AP_box on COCO, 52.7 mIoU on ADE20k). SMR does not seem to bring obvious benefits over MILAN.

[1] Hou, Zejiang, et al. "Milan: Masked image pretraining on language assisted representation." 2022.

**Questions:**

1.	Inappropriate comparison in Table 1. I suggest the authors to either remove Table 1 or compare with other methods that use CLIP as teacher. Current comparisons are unfair and can be misleading to readers.

2.	Typo in Page 5, Token compression function paragraph. TC() or CF()?

---

> ### Author Response · Authors · 2023-11-23
> **Thanks for the comments**
>
> Dear reviewer,
>
> Thanks for your time and the review. We believe that the comments will help us to further improve the quality of our work in the future. Based on the initial rating, we choose to withdraw the submission this time.
>
> Best,
>
> Authors

---

### Official Review · Reviewer_gZTV · 2023-11-01

**Soundness:** 2 fair
**Presentation:** 1 poor
**Contribution:** 2 fair
**Rating:** 3
**Confidence:** 5

**Summary:**

The paper introduces a novel visual pre-training framework called SMR, inspired by image compression theory. The experimental results demonstrate that SMR outperforms several previous visual pre-training methods in various downstream tasks, including ImageNet classification, COCO object detection, and ADE20k segmentation. The authors have also conducted extensive ablation studies to validate the design choices in SMR.

**Strengths:**

1. The paper's concept is grounded in image compression theory, providing a strong foundation for its ideas. This connection to image compression theory is a valuable angle for exploring visual pre-training.

2. The integration of semantic merging and visual pre-training is a novel approach. This direction is particularly interesting as it can significantly reduce the computational burden of models like ViT.

**Weaknesses:**

1. The paper's organization and writing quality need improvement. In the introduction, it's essential to clearly articulate the core problem that the paper aims to address. While the authors state their proposal, it's not entirely clear why this approach is needed. Additionally, the related work section lacks a coherent structure, and the relationships between this paper and the referenced works are not adequately discussed.

2. Missing some important comparisons. In the Tab.2, it is not proper to directly compare SMR with purely self-supervised method (do not use CLIP target) like MAE, MoCo, etc. In this aspect, the authors should at least emphasize this point in the table. Meanwhile, several important baselines are missing in this table, including those utilizing CLIP target (MILAN [a]) and use purely self-supervised targets (SiameseIM [b]). SMR achieves 85.4 on ImageNet with 400 epochs training which is almost the same with MILAN (85.4 with 400 epochs), thus the performance gain is marginal compared to previous methods utilizing CLIP target.

[a] Hou et al. MILAN: Masked Image Pretraining on Language Assisted Representation. ArXiv 2022.

[b] Tao et al. Siamese Image Modeling for Self-Supervised Vision Representation Learning. In CVPR 2023.

**Questions:**

Overall, the paper presents an interesting idea, but some critical aspects do not meet the typical requirements of top conferences like ICLR, as outlined in the Weaknesses section. I am open to discussing these points further.

Minor questions:

1. The claim of being "arch-free" might not be entirely accurate, as SMR is not applicable to convolution-based methods like ResNets. It's worth considering clarifying this point and distinguishing SMR from methods like MoCo and SimMIM.

2. Have you explored the possibility of discarding tokens not only during pre-training but also in the fine-tuning of ImageNet classification? This approach could lead to the development of a highly efficient network for image classification.

---

> ### Author Response · Authors · 2023-11-23
> **Thanks for the comments**
>
> Dear reviewer,
>
> Thanks for your time and the review. We believe that the comments will help us to further improve the quality of our work in the future. Based on the initial rating, we choose to withdraw the submission this time.
>
> Best,
>
> Authors

---

### Official Review · Reviewer_Zv11 · 2023-11-01

**Soundness:** 2 fair
**Presentation:** 3 good
**Contribution:** 2 fair
**Rating:** 5
**Confidence:** 4

**Summary:**

This paper presents the Semantic Merging and Reconstruction (SMR) framework, which uses image compression and reconstruction as a pretext task for vision transformer pre-training. The author argues for two major contributions of the SMR framework: explicit connections between visual pre-training and information theory, and the flexibility of the SMR framework to be applied across a wide range of different vision transformer architectures, such as vanilla ViT, HiViT, and ConViT. The main results on ImageNet-1K fine-tuning, object detection, and image segmentation are provided, demonstrating advantages over selected previous methods. Numerous ablation studies are also conducted. In summary, this paper proposes SMR, a visual pre-training method with advantages in formulation and efficiency, and also achieves good performance.

**Strengths:**

- Efficiency brought by SMR framework comapred to previous pre-training methods are definitely without any doubt. Numerical and quantitive evidences are provided by the authors in Table 1 and Table 8.
- The authors exhibit the flexibility of SMR on non-vanilla vision transformer architectures.
- The overall framework is very concise.
- Extensive ablation studies are conducted.

**Weaknesses:**

- The authors imply that their framework SMR is a self-supervised visual representation method. However, I believe SMR is more like a weakly-supervised learning due to the involvement of CLIP vision encoder. CLIP is trained by vision-language contrastive objective and the noisy text crawled from internet performs as weak supervision in CLIP's training procedure. Therefore, I would like to treat SMR framework as a weakly supervised one instead of a self-supervised framework.
- Novelty issues. The idea that leverage token reducing method (e.g., token merging, token dropping and etc.) for visual pre-training or downstream vision tasks has been explored in previous works. One of the most relevant paper to SMR is GroupViT[1], in which the authors also explored visual token merging, despite with different motivations and intentions. However, this important relevant work is neither cited nor discussed in this submission.
- Relationship between SMR and information theory. In introduction, the writers treat the relationship between their method SMR and information theory as one of the main contributions of this paper. However, the following sections do not show how the connection between information theory and SMR are drawed. Besides, as far as I am concerned, SMR has almost noting to do with information theory. Maybe SMR is more close to information bottleneck theory[2] instead of information theory.
- Scaling limitations. Most of the main results given by the authors are conducted with ViT-Base network. Then in the appendix, the authors describe their attempts to scale SMR to 10B parameters, in order to emphasize the potential and efficiency edge of SMR on further scaling up. However, it would be more persuasive and compelling if the results of SMR with ViT-L or ViT-H are given.
- Performance. Compared to the listed methods in Table 2 and Table 3, slight improvements are obtained by SMR. However, compared to other methods, the performance advantages of SMR are less significant. For instance, with CLIP ViT-B as teacher, dBoT with ViT-B network can also achieve 85.7% top-1 accuracy after fine-tuning on ImageNet-1K.

```
[1] Xu, Jiarui, et al. "Groupvit: Semantic segmentation emerges from text supervision." Proceedings of the IEEE/CVF Conference on Computer Vision and Pattern Recognition. 2022.
[2] Alemi, Alexander A., et al. "Deep variational information bottleneck." arXiv preprint arXiv:1612.00410 (2016).
[3] Liu, Xingbin, et al. "Exploring target representations for masked autoencoders." arXiv preprint arXiv:2209.03917 (2022).
```

**Questions:**

See Weaknesses.

**Details Of Ethics Concerns:**

N/A.

---

> ### Author Response · Authors · 2023-11-23
> **Thanks for the comments**
>
> Dear reviewer,
>
> Thanks for your time and the review. We believe that the comments will help us to further improve the quality of our work in the future. Based on the initial rating, we choose to withdraw the submission this time.
>
> Best,
>
> Authors